# Effects of fishing restrictions on the recovery of the endangered Saimaa ringed seal (*Pusa hispida saimensis*) population

**Pekka Jounela** [1] *, **Miina Auttila**[2], **Riikka Alakoski**[2], **Marja Niemi**[3], **Mervi Kunnasranta**[3,4]

**1** Natural Resources Institute Finland, Turku, Finland, **2** Metsähallitus, Parks & Wildlife Finland, Savonlinna, Finland, **3** Department of Environmental and Biological Sciences, University of Eastern Finland, Joensuu, Finland, **4** Natural Resources Institute Finland, Joensuu, Finland

\* pekka.jounela@luke.fi

**Data Availability Statement:** All relevant data are within the manuscript and its Supporting Information files.

## Abstract

Over the past three decades, incidental bycatch has been the single most frequent verified cause of death of the endangered Saimaa ringed seal (*Pusa hispida saimensis*). Spatial and temporal fishing closures have been enforced to mitigate bycatch, which is mainly caused by the gillnets of recreational fishers. In this study, we employed an array of statistical machine learning methods to recognize patterns of death and to evaluate the impacts of annual fishing closures (15th April–30th June) on the recovery of the Saimaa ringed seal population during 1991–2021. We additionally used the potential biological removal (PBR) procedure to assess bycatch sustainability. The study shows that gillnet restriction areas are reflected in the timing of juvenile bycatch mortality of the Saimaa ringed seal. In the 1990s, peak mortality occurred at the beginning of June, but as the restrictions expanded regionally in the 2000s, the peak shifted to the beginning of July. Longer temporal coverage of annual closures would have improved juvenile survival. The study also shows that estimated bycatch mortality is higher than observed: the estimated bycatch averaged approximately two unobserved bycatches per one observed bycatch. Despite the continuing bycatch mortality, a larger number of juveniles nowadays survive to the age of 15 months due to fishing closures, and the population (some 420 individuals) has increased an average 4% per year between 2017 and 2021. However, human-caused mortality limits (PBR) were exceeded by observed bycatch only, which could lead to population depletion in the long run.

## Introduction

The unintended capture of nontarget species, or "bycatch," is one of the main direct human-induced mortality causes of marine megafauna. Bycatch can cause injury or even death to the animals involved [1]. Marine mammals can become entangled in various fishing gear, such as trawls, purse seines, longlines, gillnets, and pot/trap fisheries, or they may be accidentally struck by fishing vessels [2–6]. Seal–fishery interactions are a major conservation concern also for several pinniped species globally [7], and incidental bycatch has been identified as a major threat for the most endangered rare seal populations [6–11].

**Funding:** This work was funded by the Ministry of Agriculture and Forestry of Finland and the Our Saimaa Seal LIFE Programme of the European Commission, grant number LIFE19-NAT-FI-000832, received by Mervi Kunnasranta (https://webgate.ec.europa.eu/life/publicWebsite/project/LIFE19-NAT-FI-000832/working-together-to-save-the-saimaa-ringed-seal-in-changing-environment). The funders had no role in study design, data collection and analysis, decision to publish, or preparation of the manuscript. The material reflects the views of the authors; neither the European Commission, the CINEA, nor the Ministry of Agriculture and Forestry of Finland is responsible for any use that may be made of the information it contains.

The landlocked Saimaa ringed seal (*Pusa hispida saimensis*) population in Finland has suffered from bycatch mortality, which has been implicated as a primary threat to the Saimaa ringed seal population [7]. After weaning, bycatch is shown to be the dominant mortality factor for juveniles [6,8], and the problem remains acute despite active conservation efforts during past decades [12,13]. The negative consequences of bycatch mortality for the population's long-term survival have been highlighted during a time in which this endangered subspecies is increasingly suffering from the changing climate [14]. Saimaa ringed seals are dependent on ice and snow as a breeding habitat and are therefore suffering from direct negative effects of climatic warming. Pups are born in subnivean lairs in February–March, and nursing lasts 7–12 weeks [15–18]. Saimaa ringed seals display high breeding site fidelity, and females are shown to exhibit at least mutual avoidance during the breeding period [19]. Perinatal mortality is relatively high and is suggested to be related to poor breeding habitat conditions due to mild winters [20] or inbreeding pressure [21]. Moulting occurs after the breeding season in May–June, and the seals use terrestrial platforms for moulting [22].

Globally, gillnets are one of the most lethal fishing gears for pinnipeds [6]. This is also seen in Lake Saimaa, where the majority of observed seal bycatch mortality is caused by gillnets [26], mainly owned by recreational fishers [8]. Only 30–40 commercial fishers work in Lake Saimaa, but the exact volume of the larger-scale recreational fishery is not known. Estimates vary between 49 000and 400 000 recreational fishers, a large proportion of which use passive gear such as gillnets and/or wire traps [23–26]. The vendace (*Coregonus albula*) is a target species of primary economic importance for commercial fishing in Lake Saimaa. It is mostly caught with seal-safer gear, such as trawls and seines, but also with gillnets in some places [27]. The other economically important target species, such as pike perch (*Sander lucioperca*), perch (*Perca fluviatilis*), and pike (*Esox lucius*), are typically caught with gillnets. These species are also the most common catches for recreational fishers [27,28]. In general, a significant part of fishing is conducted using passive gear, and 36% of all recreational fishing catches in Finland in 2020 were made using gillnets [29].

Pinniped bycatch mitigation methods are typically related to modifications of fishing gear type and practice, and to fishery closures [4,30,31]. In Lake Saimaa, these measures have included the introduction of gear modifications, such as thinner gillnet materials, fish traps, and fyke net models with bars to keep seals out, and prohibiting the use of longlines and fish-baited fishing hooks. In addition, education and outreach programmes have helped raise awareness among fishers and the public on the impacts of bycatch and the importance of responsible fishing practices [14]. The implementations of temporal and spatial fishing closures have been the key conservation measures since the 1990s [8]. The most dangerous fishing methods for seals are banned year-round in the main Saimaa ringed seal distribution area by the Government Decree on Fishing Restrictions (374/2021). In addition, springtime fishing area closures for gillnet fishing have been enforced from 15th April to 30th June, which is suggested to be the most critical period for pup survival [8]. In 1991, springtime spatial closures only covered some 18% of birth lair sites, but since the end of the 2010s this coverage has been around 95% [13]. However, impacts of the long-term temporal and spatial fishing restrictions on the population growth rate have not been evaluated, though the seal population has slowly recovered during the past decades from the strongest decline [14]. Increased measures due to seal conservation, especially fishing restrictions, have concurrently increased local socioeconomic conflicts [24].

Fishing in Lake Saimaa is not managed based on biological reference points that inform fisheries managers of the limits beyond which the state of a fishery and/or a resource is undesirable. Limits for allowable human-caused mortality can be assessed using the potential biological removal approach (PBR; [32]). In general, the PBR approach is a reference point

method originally developed to assess human-caused mortality limits of marine mammal populations [33]. The maximum net productivity level (MNPL) underpins PBR and is nearly identical to the concept of maximum sustainable yield (MSY) often used in fish stock assessments [34]. In the PBR approach for marine mammals, MNPL is often considered to be 50% or more of the environment's carrying capacity (K), and the optimum sustainable population (OSP) is defined as being between K and MNLP. Under the U.S. Marine Mammal Protection Act [35], the law for which the PBR approach was developed, populations should not be permitted to diminish below the OSP level. If this happens, management actions may be taken to recover the depleted population. The PBR approach provides a reference point and management process that have been shown in simulations to have a high probability of maintaining a stock within the OSP range, even when population data are uncertain [36]. Estimating bycatch limits in terms of PBR reference points would also improve conservation planning regarding the Saimaa ringed seal population.

We present the key findings of an analysis conducted to estimate the bycatch mortality of the Saimaa ringed seal caused by fishing for 1991–2021. The aim is to evaluate the impact of the past and proposed extended temporal fishing restrictions on the recovery of the seal population. The specific objectives are 1) to estimate the timing of the bycatch mortality related to the spatial coverage of the fishing closures, 2) to study how the extended temporal coverage of annual restrictions would have improved pup survival, 3) to estimate the magnitude of unobserved bycatch mortality, 4) to estimate population size, and 5) to use the PBR approach for calculating whether observed and estimated bycatch levels are consistent with reaching the target growth rate of 3–6% per year using a five-year moving average to reduce the impact of random and natural annual variation [37]. We employed an age group-specific population model and an array of machine learning techniques to recognize patterns of deaths that are too complex to be perceptible by human researchers employing standard statistical tools.

## Materials and methods

### Study area

Lake Saimaa (61˚ 05−62˚ 36' N, 27˚ 15−30˚ 00' E) is a labyrinthine freshwater system in eastern Finland. It is approximately 180 km long and 140 km wide, and the study area covers the Saimaa ringed seal distribution (Fig 1). Metsähallitus Parks & Wildlife Finland (MHPWF) is a public authority responsible for Saimaa ringed seal population monitoring and most of the conservation actions. Data on seal mortality, lair sites, and numbers of pups born in 1991−2021 used in the analyses were obtained from the MHPWF database [37]. In addition, a spatial coverage estimate (%) of the springtime (15th April–30th June) fishing area closures for gillnet fishing in relation to birth lairs was used (i.e. birth site locations % inside fishing restriction area, see also [13]).

### Data on mortality

Since the 1990s, all dead Saimaa ringed seals that have been reported to MHPWF have been collected and examined. The mortality data used in our study, including causes of death (Table 1), location, date, and age, were based on 554 carcasses examined between 1991 and 2021. The cementum layers in the lower canine teeth were counted to determine seal ages (apart from pups with lanugo hair; [38]). In addition, 98 individuals in the dataset were not included in this study due to lack of information on their exact age. A wildlife pathologist identified the causes of death, or field verification was obtained for instances of entanglement in fishing gear. In addition to those known to have died as bycatch (found in fishing gear), other carcasses determined by a wildlife pathologist to have drowned were also defined as observed

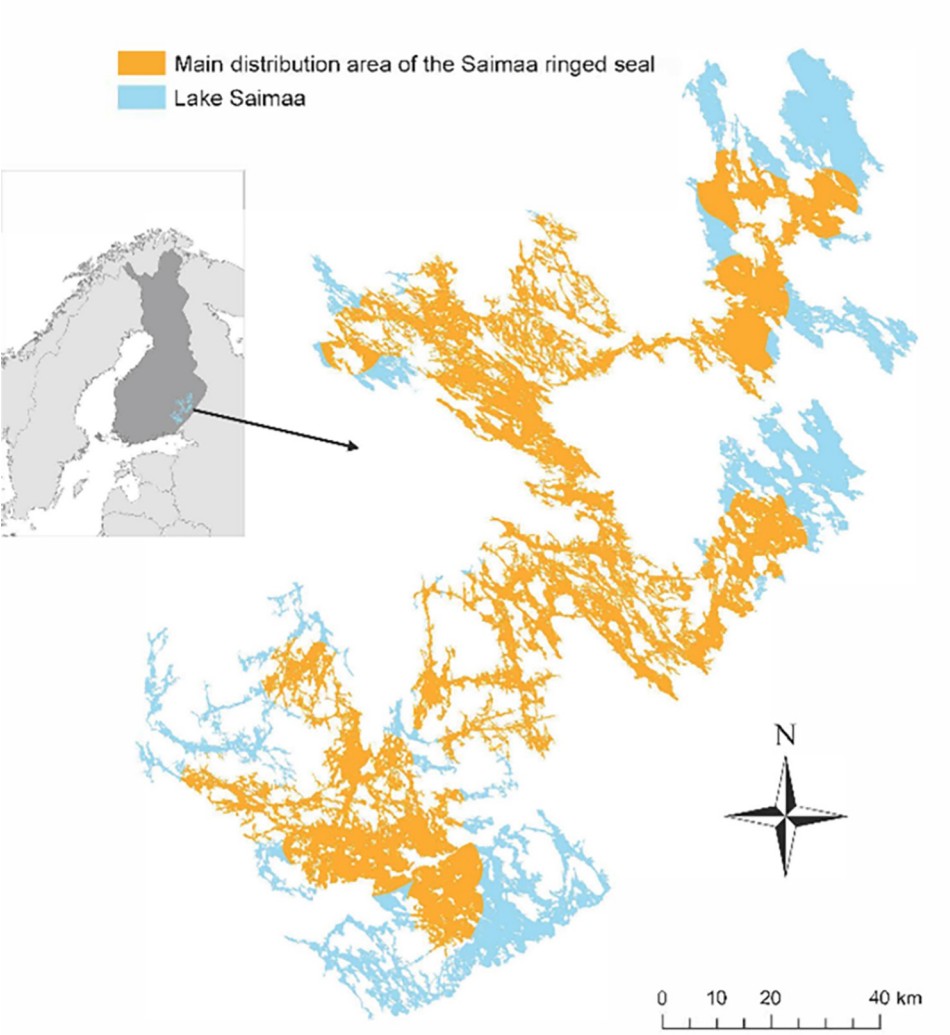

**Fig 1. Lake Saimaa and the current main distribution area of the Saimaa ringed seal.** Data from the Finnish National Land Survey 1:100 000 geographical map (downloaded 1/2015), MHPWF Saimaa ringed seal breeding lair sites in 2010–2022, and https://www.naturalearthdata.com/downloads/10m-cultural-vectors/10m-admin-0-countries/ (downloaded 2/2023).

bycatch mortality (Table 1). Hereafter, we define the word "observed" as information that exists in the MHPWF database as opposed to "unobserved" information that does not exist in any database. A model result is referred to as "estimated", and unobserved information is the difference between estimated and observed information (i.e. estimated = observed + unobserved).

## Data on pups and birth lairs

As Saimaa ringed seals give birth to a single pup, birth lairs are used to estimate the annual number of pups born since the 1990s [8]. Information on lair sites and pup numbers are based on lair censuses and on diving surveys of lair sites from 2014 onwards, conducted annually throughout Lake Saimaa [20,39]. Observed pup numbers are based on confirmed evidence of birth (placenta, lanugo hair, alive or dead pup). However, not all pups are found with absolute

**Table 1. Causes of death frequencies and four main cause of death groups in the Saimaa ringed seal mortality data.**

| Cause of death, all age groups | Frequency | Four main cause of death groups | Frequency |
|---|---|---|---|
| Stillborn | 63 | Stillborn | 63 |
| Predated lair deaths | 5 | Lair deaths | 108 |
| Other lair deaths | 97 | | |
| Unweaned other deaths | 6 | | |
| Fishing (self-reported) | 142 | Fishing | 163 |
| Suffocation | 21 | | |
| Natural | 26 | Other | 220 |
| Unknown | 186 | | |
| Accidental | 1 | | |
| Cancer | 2 | | |
| Boat collision | 4 | | |
| Violence | 1 | | |
| Sum | 554 | | 554 |

certainty, which is why the number of born pups is estimated using a three-year average in areas where the census has not been comprehensive during the current year. Therefore, birth lair data included both observed and estimated numbers of pups and locations of birth lairs during 1991−2021.

## Statistical methods

**Model structure.** Bycatch was estimated using an age group-specific population model in Rapidminer Studio Enterprise software (version 9.10.011; www.rapidminer.com, [40], see also [13]). The Rapidminer software is based on the eXtensible Markup Language (xml) that also supports open-source languages, such as Python and R in an integrated fashion. The advantage is that one can thereby run an extremely wide range of parametric and non-parametric (sub-) models in a single platform. We analysed seal survival between 1991 and 2021 as two age groups: 'pups' from ages 0 to 15 months, and 'adults' up to 30 years of age in 1-year intervals. The total mortality rate, i.e. the number of all deaths at age at x, was computed separately for adults and pups. This two-stage approach was used because most carcasses (68%) were of individuals less than 15 months old [13], and for this reason the annual change in the adult population size was estimated starting with the survivors in the 15-month age group. Due to the low annual number of samples in the adult carcass data (5.4 individuals per year on average), adult seals were pooled in a joint survival estimate for 1991–2021. The population size estimate included 30 age groups (from 1 year to 30 years). The starting assumption was that the 1990 population size of this study equalled that of MHPWF (mean 189, range 164–210; [26]).

The starting data used in the analysis was the total number of annually born pups minus the number of observed deaths at age 0 months (*Nage0*). However, not all deaths are observed during the first two-week period, and hence the total number of 0.5-month-old seals (*Nage0.5*) at the end of the first two-week period (0 -> 0.5 months) was estimated as:

$$N_{age0,5} = N_{age0} - N_{age0} * exp^{\left(-\frac{b}{2}\right)} * Pr_{age0} \tag{1}$$

where *b* is the slope parameter of the linear regression line directly proportional to the total mortality rate assessed by plotting the carcass frequencies (ln + 1 transformed) by month and year of age (also known as the slope parameter of the 'catch curve' mortality estimation method; [41,42], 2 denotes catch curve mortality estimation in 2-week age periods, and Pr

denotes the k-nearest neighbour model (k-NN; [43]) -based proportion (range: 0, . . ., 1) of deaths during the first 2 weeks of age. Parameter k of the k-NN model was optimized using 10-fold cross validation [44]. The catch curve mortality estimation method infers total mortality rates based on the changes in (carcass) age distribution. It is a widely used technique in fisheries science, especially with data-poor fish stock assessments. The catch curve mortality estimation method assumes that mortality is the only reason for the decrease in individual numbers across the ages of the population i.e., there is no emigration or immigration in the study area. Note here that the catch curve mortality estimation method assumes age-constant mortality rates, and hence we also apply nonlinear k-NN-based proportions of deaths in 2-week periods to recognize more realistic nonlinear mortality patterns. Bycatch has never been observed during the first 2-week age period and was hence assumed to be zero. Thereafter, bycatch C at age 0.5–15 months during the closed and open gillnet seasons and area was calculated as:

$$C_{age} = N_{age} - N_{age} * exp^{\left(-\frac{k}{2}\right)} * Pr_{age} * P(C_{age}) * O_{age} \tag{2}$$

where *P(Cage)* denotes confidence (sometimes also referred to as probability; range 0, . . ., 1) of bycatch death with respect to all other causes of deaths at age 0.5–15 months and *Oage* denotes the proportion (range 0, . . ., 1) of open gillnet fishing area at age. The remaining deaths at age 0.5–15 months caused by other reasons occurred in all (open and closed) areas. Alternative approach to estimate abundance, trend and population viability of an isolated small seal population is presented in [45] which used Bayesian hierarchical models that were developed from aerial survey and harvest data. In addition, [46] developed a Bayesian integrated population model that combined mark-recapture and count data to estimate demographic rates, abundance trends, and the effects of environmental variability on population growth of endangered Steller sea lions. In general, the pros of using Bayesian methods are the probabilistic framework and inclusion of prior information whereas the goal of this study was mostly to identify patterns of mortality that can be too hazy for human researchers to notice.

**Causes of death estimation.** The confidence distributions for the causes of death ('distribution of proportions among categories') in terms of age, year, and water area were computed using four main cause of death groups: stillborn, which consisted of pups dead at birth; lair death, which consisted of mostly unverified causes of death of unweaned pups aged 0–2 months (multiple, mostly unverified causes of death); fishing, which consisted of observed bycaught seals aged 0–35 years; and other, which consisted of seals older than 2 months that died of natural or unknown causes, of which some might have died from fishery interactions.

The reason for not using all the original cause of death groups (N = 12, Table 1) was to minimize the computational complexity of the algorithm. On the other hand, more than two cause of death groups (fishing and other than fishing) were used in the model because our trial model runs showed better 10-fold cross-validation performance of the model. Cause of death patterns were recognized using a random forest (RF) model [47]. In general, RF is a machine learning method that consists of building an ensemble [48] classifier of decision trees produced from bagging (bootstrap aggregation; see [49]) and a randomized variant of the tree induction algorithm. The four parameters of the RF model, i.e. number of trees, maximal depth, criterion (accuracy, gini index, information gain, gain ratio), and confidence level used for the pessimistic error calculation of pruning, were optimized using 10-fold cross-validation [43].

**Simulated change in fishing restrictions.** We also simulated the response in survival rate if (and only if) the start of gillnet fishing in 1991–2021 had been the 1st of August, the 1st of October, or the ban on gillnet fishing had been year-round. Currently, fishing restrictions

expire at the end of June. There is ongoing debate on continuing these restrictions and on using month as the temporal unit. We therefore also use months as an example. These responses in survival rate may be transitory because they do not consider the possible long-term effects after the change, such as a possible increase in recruitment in upcoming years.

**Potential biological removal.** The PBR approach [32] was used for calculating whether observed and estimated bycatch levels ensure a high probability that the population will trend toward or stay within the OSP range of abundance. The PBR was calculated as:

$$PBR = N_{Min} \frac{1}{2} R_{Max} \, F_R \tag{3}$$

where NMIN is the 20th percentile of a log-normal distribution based on an estimate of the number of animals in a stock (which is equivalent to the lower limit of a 60% two-tailed confidence interval (CI)), ½ RMAX is one-half of the maximum theoretical or estimated net productivity rate of the stock at a small population size, and FR is a recovery factor between 0.1 and 1. The lower 60% CI bound concept in PBR was used to derive Nmin from the best estimate of N. The input values used here were default ones used with endangered pinniped populations RMAX = 0.12 and FR = 0.1 [50].

## Results

In 1991–2021, fishing was responsible for 26% (n = 142 / 554; Table 1) of the observed (self-reported) mortality and 90% (n = 128) of these bycaught seals were entangled in gillnets.

### Temporal distribution of bycatch mortality

The distribution of estimated bycatch mortality during the first 15 months of a Saimaa ringed seal's life showed two peaks over the past thirty years (Fig 2A). The estimated first bycatch peak in the 1990s occurred at the beginning of June, when pups were three months old. Since the early 2000s, the first estimated bycatch peak shifted to the beginning of July (Fig 2A). However, temporal shifts did not occur in the bycatch peak at the beginning of March, when the ringed seals born the previous year were approximately one year of age.

### Delayed start date of gillnet fishing

A longer temporal closure for gillnet fishing would improve juvenile survival. Postponing the start of gillnet fishing to early August would have increased juvenile survival (0–15 months) by 1.8 individuals in 2021 (Figs 2B and 3, S1 Fig and S1 Table). Similar, but possibly transitory, estimates for gillnet fishing starting in early October and a year-round ban on gillnet fishing in 2021 would have increased survival by 4.5 and 13.1 individuals, respectively (Figs 2C, 2D and 3). However, postponing the start of gillnet fishing to the beginning of August or October would have increased the bycatch numbers of winter fishing (Fig 2C and 2D). Nevertheless, winter and autumn fishing would not eliminate the benefit (increase in survival) that would result from postponing the start of gillnet fishing by 1–3 months (Figs 2A–2D and 3). Note that the estimated increases in survival are instantaneous estimates, which means that the calculation does not consider the possible long-term (multi-year) cumulative effects of the change.

### Estimated bycatch numbers

The bycatch estimated in this study averaged 1.8 unobserved fishing deaths per one observed. The number of observed fishing-related deaths has increased somewhat compared to the

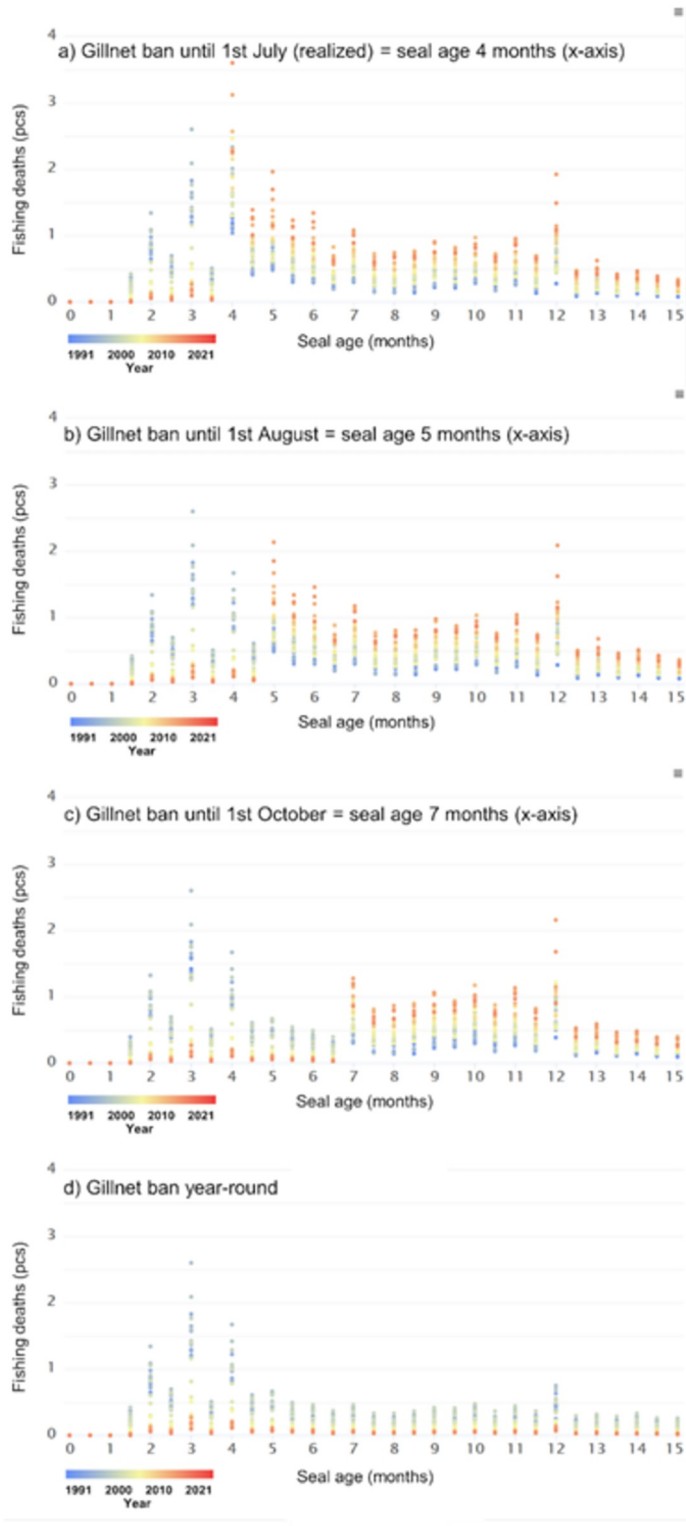

**Fig 2.** Estimated bycatch of Saimaa ringed seal juveniles (0 to 15 months) between 1991 and 2021: (a) the actual start of gillnet fishing, with the date 1st July corresponding to a ringed seal age of four months, b) the start of gillnet fishing, with the date 1st August corresponding to a ringed seal age of five months, c) the start of gillnet fishing, with the date 1st October corresponding to a ringed seal age of seven months, and d) the year-round ban of gillnet fishing. For example, the first bycatch peak (y-axis) in the 1990s (blue colour) (Fig a) occurred in June due to the small coverage of

closed areas (x-axis), after which the first bycatch peak (y-axis) moved forward to the beginning of July (x-axis), as the area closed to gillnet fishing increased from the beginning of the 2000s (yellowish–reddish colour). In 1991, closed areas covered approximately 8% and, since the end of the 2010s, closed area coverage was approximately 95% of juvenile distribution areas.

1990s. The observed annual bycatch averaged 4.44 individuals in the 1990s, 6 individuals in the 2000s, and 5.25 individuals in 2010–2021 (Table 2). Over the same period, the number of bycatches estimated by the model, i.e., the number of observed and unobserved deaths, has increased by 1.7-fold and the number of ringed seals that survived the first 15 months has increased by 3.1-fold. For example, the average annual estimated number of bycaught seals in 2010–2021 was 18.86 (Fig 4). Seal bycatch rate in terms of average annual estimated number of bycaught seals per average annual estimated number of survivors at age 0.5 months was 0.41 in the 1990s, 0.37 in the 2000s, and 0.32 in 2010–2021 (max 0.46 in 1993; min 0.24 in 2021; Table 2). The carcass data suggested an average 40% of the carcasses >0.5 months old that were found independent of fishing gear entanglements were determined to have died by suffocation (drowning) in 1991–2021. During the past decade, an annual average of ~ 70 survivors in the over 0.5-month-old recruits and an increase in the population size by ~ 10 individuals suggested ~ 60 deaths, of which roughly 24 (60 x 40% = 24) individuals had suffocated (bycatch). In addition, during 1991–2000, an annual average of 33 estimated survivors in the

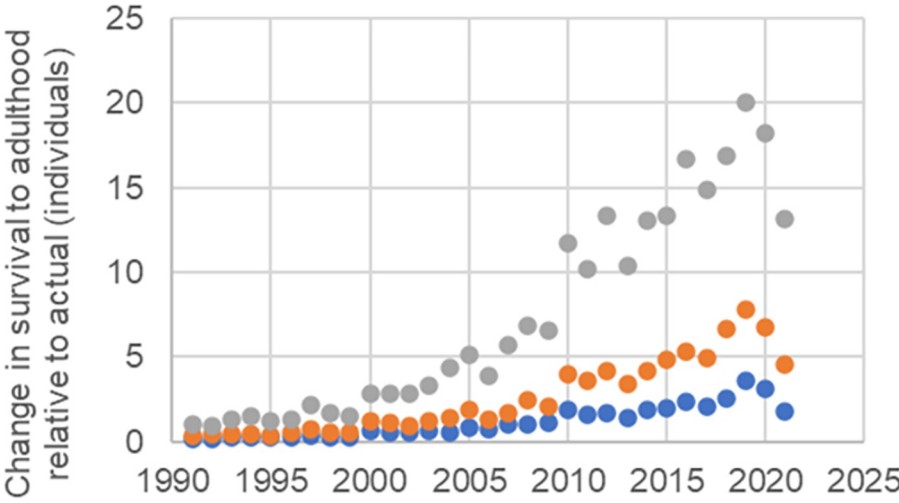

**Fig 3. Change in juvenile survival to adulthood relative to actual survival (number of surviving individuals, y-axis) if temporal closures would have begun on 1st August (= 1-month delay; age of ringed seal five months), 1st October (= 3-month delay; age of ringed seal seven months), or if the temporal closure had been year-round from 1991 to 2021 (x-axis).** For example, the interpretation of the blue dot on the right: Moving the start of gillnet fishing forward by one month to the beginning of August (blue colour) in 2021 (x-axis) would have increased the survival of 0–15-month-olds by 1.79 individuals (y-axis). Increases in survival are instantaneous estimates, which means that the calculation does not account for the possible long-term cumulative effects of the change.

**Table 2. Estimated and observed bycatch, number of pups born, estimated all deaths, and estimated other deaths of seals aged 0–15 months (see also S1 Table), estimated 15-month-old survivors, estimated population size (mean; number of individuals), and bycatch rate of seals in terms of the average annual estimated number of bycaught seals/average annual estimated number of survivors at age 0.5 months in 1991–2021.** During 2017–2021, the population growth rate averaged 4%. In total, birth lair data included the observed and estimated numbers of pups (observed N = 1557, estimated N = 1778) and birth lair locations (N = 1526) during 1991–2021.

| Year | Est. bycatch | Obs. bycatch | Est. no. of pups born | Est. all deaths | Est. other deaths | Est. 15-month-old survivors | Est. population size | Est. bycatch / Est. 0.5-month-old survivors |
|---|---|---|---|---|---|---|---|---|
| 1991 | 9.7 | 3 | 31 | 16.6 | 6.8 | 11.4 | 189.7 | 0.44 |
| 1992 | 9.7 | 1 | 34 | 18.3 | 8.6 | 12.7 | 191.0 | 0.39 |
| 1993 | 13.0 | 9 | 36 | 20.0 | 7.0 | 14.0 | 193.5 | 0.46 |
| 1994 | 11.4 | 4 | 38 | 18.2 | 6.8 | 12.8 | 194.7 | 0.44 |
| 1995 | 10.4 | 4 | 39 | 21.2 | 10.9 | 14.8 | 197.8 | 0.36 |
| 1996 | 9.5 | 0 | 40 | 15.8 | 6.3 | 11.2 | 197.2 | 0.44 |
| 1997 | 15.7 | 7 | 44 | 24.6 | 8.9 | 17.4 | 202.9 | 0.45 |
| 1998 | 11.9 | 10 | 41 | 22.2 | 10.3 | 15.8 | 206.7 | 0.37 |
| 1999 | 10.7 | 2 | 42 | 23.7 | 13.0 | 16.3 | 210.8 | 0.34 |
| 2000 | 17.6 | 6 | 57 | 31.5 | 13.9 | 22.5 | 221.0 | 0.42 |
| 2001 | 16.3 | 4 | 56 | 27.4 | 11.1 | 19.6 | 227.8 | 0.43 |
| 2002 | 14.8 | 8 | 56 | 31.3 | 16.5 | 22.7 | 237.4 | 0.33 |
| 2003 | 16.2 | 8 | 51 | 27.1 | 10.9 | 19.9 | 243.7 | 0.42 |
| 2004 | 18.6 | 6 | 66 | 35.0 | 16.4 | 25.0 | 254.9 | 0.36 |
| 2005 | 15.7 | 5 | 51 | 26.7 | 11.0 | 20.3 | 260.8 | 0.41 |
| 2006 | 12.7 | 6 | 52 | 23.2 | 10.6 | 17.8 | 264.0 | 0.37 |
| 2007 | 12.7 | 6 | 55 | 26.1 | 13.4 | 20.9 | 270.1 | 0.32 |
| 2008 | 11.9 | 5 | 51 | 26.1 | 14.2 | 20.9 | 276.0 | 0.30 |
| 2009 | 9.5 | 6 | 44 | 20.5 | 11.0 | 17.5 | 278.2 | 0.31 |
| 2010 | 16.0 | 5 | 57 | 28.4 | 12.4 | 24.6 | 287.4 | 0.37 |
| 2011 | 13.0 | 4 | 57 | 24.8 | 11.8 | 21.2 | 292.8 | 0.34 |
| 2012 | 16.9 | 7 | 61 | 31.5 | 14.6 | 26.5 | 303.3 | 0.35 |
| 2013 | 13.7 | 7 | 62 | 27.9 | 14.2 | 23.1 | 309.9 | 0.31 |
| 2014 | 15.4 | 7 | 64 | 29.6 | 14.1 | 25.4 | 318.5 | 0.33 |
| 2015 | 15.7 | 5 | 71 | 33.7 | 18.0 | 28.3 | 329.6 | 0.31 |
| 2016 | 20.1 | 5 | 87 | 43.4 | 23.3 | 35.6 | 347.5 | 0.30 |
| 2017 | 17.7 | 7 | 83 | 37.3 | 19.6 | 30.7 | 359.7 | 0.31 |
| 2018 | 19.7 | 4 | 86 | 42.0 | 22.3 | 34.0 | 374.7 | 0.31 |
| 2019 | 22.7 | 6 | 88 | 46.8 | 24.1 | 40.2 | 395.2 | 0.33 |
| 2020 | 20.7 | 4 | 88 | 42.5 | 21.8 | 34.5 | 409.0 | 0.33 |
| 2021 | 15.9 | 2 | 90 | 45.2 | 29.3 | 33.8 | 421.6 | 0.24 |

over 0.5-month-old recruits and an estimated increase in population size by 4 individuals suggested an estimated 29 deaths, of which 11.6 (29 x 40% = 11.6) individuals had suffocated (bycatch) (see also Table 2 and Fig 4).

## PBR

The estimated bycatch in 2021 (15.88 individuals) was higher than the PBR estimate (2.48) for that year. In most (27/31) years the observed and in all years the estimated bycatch was higher than the estimated PBR in 2021 (Table 2). The average estimated bycatch in 1991–2021 (14.69 individuals) was 5.9 times higher than the PBR in 2021. The lower limit of a 60% two-tailed confidence interval for stock size (NLower60%) in 2021 was 414 individuals.

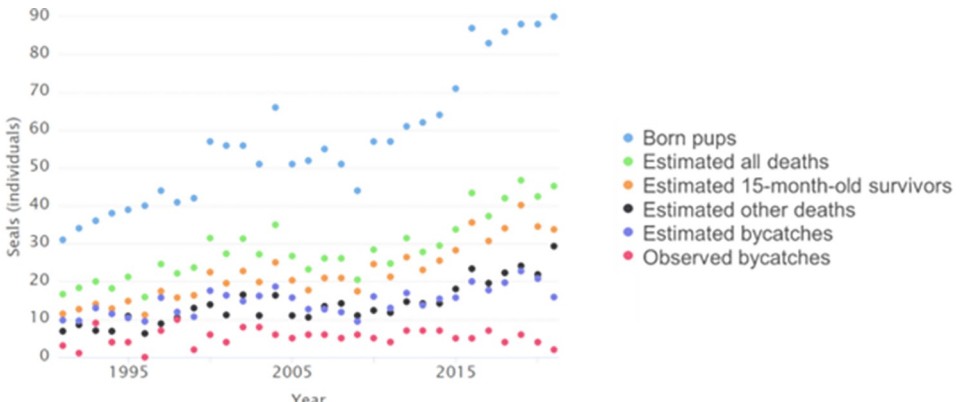

**Fig 4. Number of born Saimaa ringed seal pups, estimated all deaths of max 15-month-old seals, estimated 15-month-old survivors, estimated other deaths, estimated bycatches, and observed bycatches (individuals) between 1991 and 2021.**

## Discussion

This is the first study that investigates the effects of fishing restrictions on the endangered Saimaa ringed population over three decades. Based on our results, bycatch mortality seems to be up to three times more frequent than what is observed, especially in young seals up to 15 months of age. This creates an increased depletion risk for the population, as the present bycatch mortality rates exceed the estimated allowable human-caused mortality.

### Bycatch distribution

Saimaa ringed seals are most vulnerable to bycatch mortality as juveniles, and 90% of lethal entanglements are caused by gillnets. Bycatch typically occurs in summer, but this mortality can also be seen during other seasons. Over the previous 30 years, the bycatch mortality distribution during the first 15 months of a Saimaa ringed seal's life has been two-peaked. The first peak occurred during June–July and the second peak in March. At the time of minor fishing restrictions in the 1990s, the bycatch mortality peak occurred already at the beginning of June, but as the restrictions expanded regionally in the 2000s, the peak shifted to the beginning of July. The second mortality peak in March is caused by stationary gillnets set under ice. This is somewhat contrary to the raw data, where bycatch mortality was quite similar throughout January–March. The difference may have been caused by the relatively small number of bycatches or by parametrization of the RF model. Nonetheless, wintertime gillnetting causes mortalities because it has not been restricted in most areas. Our study indicates that the coverage of present springtime closures in the main Saimaa ringed seal distribution area may be regionally adequate, but temporally too short. Therefore, a longer temporal closure for gillnet fishing would be an important conservation action to improve juvenile survival. For example, extending fishing restrictions until the end of July in 2021 would have saved around two seals (1.79) and a ban until the end of September around four (4.5) seals. In addition, autumn and winter fishing would not eliminate the increase in survival if gillnet fishing were postponed further until the end of July or September. Also, a total ban on gillnet fishing in Lake Saimaa has been publicly discussed, and a year-round gillnet closure would have increased survival by 13 seals in 2021.

### Unobserved bycatch

Most Saimaa ringed seal bycatch mortality is unobserved. Our observed bycatch mortality was roughly one-third of the estimated fishing deaths. It is widely accepted that accurate estimation

of bycatch rates requires an independent observer scheme [51]. Observed bycatch typically vary, and they do not represent the actual magnitude of bycatch mortality [e.g., 4,52]. Underreporting may also be unintentional, as small entangled marine mammals (such as seals) may free themselves from the fishing gear while it is being retrieved, unseen by fishers [53]. Bycatch reporting for seals has been mandated in Lake Saimaa (62 §, Fishing Act 379/2015; Finnish legislation available at www.finlex.fi). Reporting rates are frequently low, and using these data usually leads to negatively biased bycatch rate estimates [54–56]. In the present study, the average estimated bycatch level was based on both fisher-reported and fishery-independent drowned carcass recoveries.

The estimated bycatch of the present study could be biased because it is statistically confounded with estimated other (natural and unknown) deaths of over 0.5-month-old seals. That is, estimated other mortality factors affect the estimated bycatch, and vice versa. Another potential bias source is the inability to reliably quantify the latent causes of bycatch, such as sudden changes in fishing effort. Overall, we have little information on the fishing effort in Lake Saimaa, and the incidental bycatch reporting rate may vary between seasons and years.

## Depletion risk

Although the Saimaa ringed seal population is slowly growing, it is noteworthy that the small population has an extremely narrow gene pool [21,57,58], and it is seriously threatened by habitat loss due to human activities and climate change [14,20,59]. All these factors increase population vulnerability to decline when anthropogenic influences, such as bycatch, increase total mortality levels. Accordingly, our PBR results suggest an increased risk of depletion and therefore conservation actions, e.g., longer gillnet closures, should be implemented. In most years, even the observed bycatch was higher than the estimated PBR. In addition, when estimated bycatch numbers are included, annual human-induced mortality was 5.9 times larger than the estimated PBR in 2021. According to Wade [32], a marine mammal population is at risk of going extinct if the human-caused mortality estimate exceeds its PBR. Even if the extra conservative Fr value 0.1 were not selected and an Fr of 0.5 were selected instead, the PBR for Saimaa seals would still be exceeded.

While habitat availability and quality are reduced by other indirect causes, bycatch influences population trends. Unintentional bycatch directly increases the risk of extinction. There are inherent delays in detection of a decline in abundance and management responses to a decline. The issue becomes more severe for smaller populations, as the accuracy of estimating their abundance decreases along with their population size [60]. Consequently, a small population experiencing a decline could become functionally extinct before any significant decline is detected [32]. Therefore, although the Saimaa ringed seal population has been growing during 2017–2021 by an average 4% annually due to active measures, bycatch mortality is an ongoing risk for the population's long-term survival.

## Conclusions

Globally, incidental bycatch in marine fisheries is typically related to commercial fisheries [e.g., 31,50,61]. However, the ecological impacts of small-scale fisheries can be severe for many species groups [e.g., 62–64]. In Lake Saimaa, recreational fishing is the major cause for ringed seal bycatch mortality. The gillnet has, by far, been the most lethal gear type for ringed seals, accounting for 90% incidental bycatch. The lethality of gillnets has also been shown in the Baltic Sea [58] and worldwide [4]. Also, other endangered species are bycaught in gillnets in Lake Saimaa, such as the landlocked Atlantic salmon (*Salmo salar m. sebago*), the Arctic charr

(*Salvelinus alpinus*), and the wild trout (*Salmo trutta*). For these reasons, more effective gillnet fishing restrictions are needed in Lake Saimaa.

## Supporting information

**S1 Fig. Estimated temporal distribution of all other deaths (other than fishing) of Saimaa ringed seals aged 0–15 months in 1991–2021.**
(TIF)

**S1 Table. Estimates of Fig 3.**
(DOCX)

**S1 Data.**
(XLSX)

## Acknowledgments

We thank wildlife pathologist Marja Isomursu (Finnish Food Safety Authority) for performing the seal carcass necropsies and Mairi Young for comments on the manuscript.

## Author Contributions

**Conceptualization:** Pekka Jounela, Miina Auttila, Riikka Alakoski, Marja Niemi, Mervi Kunnasranta.

**Data curation:** Pekka Jounela, Miina Auttila, Riikka Alakoski, Marja Niemi, Mervi Kunnasranta.

**Formal analysis:** Pekka Jounela, Miina Auttila, Riikka Alakoski, Marja Niemi, Mervi Kunnasranta.

**Funding acquisition:** Pekka Jounela, Miina Auttila, Riikka Alakoski, Marja Niemi, Mervi Kunnasranta.

**Investigation:** Pekka Jounela, Miina Auttila, Riikka Alakoski, Marja Niemi, Mervi Kunnasranta.

**Methodology:** Pekka Jounela, Miina Auttila, Riikka Alakoski, Marja Niemi, Mervi Kunnasranta.

**Project administration:** Pekka Jounela, Miina Auttila, Riikka Alakoski, Marja Niemi, Mervi Kunnasranta.

**Resources:** Pekka Jounela, Miina Auttila, Riikka Alakoski, Marja Niemi, Mervi Kunnasranta.

**Software:** Pekka Jounela.

**Supervision:** Pekka Jounela, Miina Auttila, Riikka Alakoski, Marja Niemi, Mervi Kunnasranta.

**Validation:** Pekka Jounela, Miina Auttila, Riikka Alakoski, Marja Niemi, Mervi Kunnasranta.

**Visualization:** Pekka Jounela, Miina Auttila, Riikka Alakoski, Marja Niemi, Mervi Kunnasranta.

**Writing – original draft:** Pekka Jounela, Miina Auttila, Riikka Alakoski, Marja Niemi, Mervi Kunnasranta.

**Writing – review & editing:** Pekka Jounela, Miina Auttila, Riikka Alakoski, Marja Niemi, Mervi Kunnasranta.

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
