## [Decision Letter · Decision Letter 0]

5 Aug 2024

PONE-D-24-24142Effects of fishing restrictions on the recovery of the endangered Saimaa ringed seal (Pusa hispida saimensis) populationPLOS ONE

Dear Dr. Jounela,

Thank you for submitting your manuscript to PLOS ONE. I have now received reviews from two marine mammal specialists who have disclosed their identities in completing their reviews. Both Brendan Kelly and Peter Boveng have returned constructive reviews that I think should be helpful for improving the quality of your contribution. Please consider their suggestions, particularly with respect to Peter Boveng's recommendations regarding population modeling approaches and the marked up editing suggestions that both reviewers provided. In part because the suggestions are quite clear, I have categorized this as a minor revision that we are requesting here, although Peter Boveng's review ticked the major revision category. In any case, while the manuscript does not fully meet PLOS ONE’s publication criteria as it currently stands, I am pleased to extend this invitation to revise the manuscript, which I judge to be a significant contribution of knowledge that can be applied towards protection of the Saimaa ringed seal.

We look forward to receiving your revised manuscript.

Kind regards,

Lee W Cooper, Ph.D.

Section Editor

PLOS ONE

Journal Requirements:

3. Thank you for stating the following financial disclosure: "This work was funded by the Ministry of Agriculture and Forestry of Finland and the Our Saimaa Seal LIFE Programme of the European Commission (LIFE19NAT/FI/000832)."

4. In the online submission form, you indicated that "The data is available upon reasonable request."

Reviewers' comments:

Reviewer's Responses to Questions

**Comments to the Author**

1. Is the manuscript technically sound, and do the data support the conclusions?

Reviewer #1: Yes

Reviewer #2: Yes

2. Has the statistical analysis been performed appropriately and rigorously? 

Reviewer #1: Yes

Reviewer #2: I Don't Know

3. Have the authors made all data underlying the findings in their manuscript fully available?

Reviewer #1: Yes

Reviewer #2: No

4. Is the manuscript presented in an intelligible fashion and written in standard English?

Reviewer #1: Yes

Reviewer #2: Yes

5. Review Comments to the Author

Reviewer #1: This work rigorously uses the potential biological removal approach to forecasting the consequences for Saimaa seals of human "takes." The analysis is quite thorough, and it is good that you showed that relaxing the recovery factor still indicates unsustainable fisheries takes. Similarly, I appreciated that you emphasized the latency of detecting decline and its conservation implications.

The description of methods and the data table make it clear the basis for the analysis. Figure 2, however, is difficult to read, and I recommend removing it. Figure 3 makes clear the important information.

In the attached pdf, I have suggested additional edits, mostly to clarify sentences.

Reviewer #2: Review of PONE-D-24-2142

Summary of topic and its importance

This study addresses a substantial threat to Saimaa seals, one of the most endangered seals in the world. Bycatch mortality in recreational fisheries is the dominant mortality factor for juvenile Saimaa seals. A time series of observed mortalities, categorized by causes of death, is fit in a fisheries-style ‘catch curve’ analysis, along with data on births, to estimate the annual bycatch mortality and population size. The model is used to investigate the effects of various time-area closures for the fisheries, demonstrating that extending the current closures or making them year-round would substantially increase juvenile survival and reduce the risk of population depletion.

The topic is important in Finland for contributing information to help evaluate controversial modifications to fishery closures. It is important more broadly because there is international attention on conservation of Saimaa ringed seals. For examples, the U.S. lists the subspecies as endangered, and the European Union’s Habitats Directive lists the Saimaa seal as a priority for protection and imposes limits on human activities that pose threats.

While I believe the study has produced useful results, I think the following issues warrant consideration:

ML vs model-based approach

The analysis is heavy on machine-learning techniques and non-parametric approaches. These are not weaknesses, in themselves, but they can be difficult to relate to underlying processes. These approaches also may not produce realistic distributions for the uncertainty in the results. I’m unable to assess whether these are significant issues in this case, partly because of my relative lack of experience with these methods, but also because the model fitting seems to have been done with RapidMiner, a data mining package that I had not heard of or seen used for fitting models to wildlife demographic data. RapidMiner is not open source and doesn’t appear to be free for an institutional user, so I was not able to learn much more about it.

Although the overall ‘catch-curve’ approach seems reasonable, I can’t really tell what’s going on ‘under the hood’. For me, and I suspect also for most readers of this paper, it would be clearer and more relatable to pose this as an integrated population model (IPM) structured in a hierarchical way that propagates uncertainty from inputs to results in a rational (probabilistic) and transparent way. A couple of examples that come to mind are Warlick et al. (2023) and Boveng et al. (2018); the latter has many similarities to the present study (small population of freshwater seals, time series of pup production, 2-stage demographic model, human-caused mortality as an input). I’m not saying the analysis necessarily needs to be redone, but I think there should at least be some mention of alternative approaches and consideration of the pros and cons of the approach used here.

Boveng, P.L., Ver Hoef, J.M., Withrow, D.E., and London, J.M. 2018. A Bayesian analysis of abundance, trend, and population viability for harbor seals in Iliamna Lake, Alaska. Risk Analysis 38:1988-2009. doi:10.1111/risa.12988

Warlick AJ, Johnson DS, Sweeney KL, Gelatt TS, Converse SJ (2023) Examining the effect of environmental variability on the viability of endangered Steller sea lions using an integrated population model. Endang Species Res 52:343-361. https://doi.org/10.3354/esr01282

Uncertainty in population estimates

The study uses the U.S. marine mammal management concept of potential biological removal (PBR) to add perspective about the results of the bycatch simulations. The PBR approach requires a minimum estimate of the population size that is specified as a percentile of the probability distribution for the estimates. To provide the specified performance as a management control rule, this probability distribution should reflect the full uncertainty in the population estimate, including both process and observational uncertainty. For reasons that I mentioned above, it’s not clear to me that the ML approach yields a realistic estimate of uncertainty in the population estimate. The 20th %-tile (414 seals) differs by only 7 individuals from the overall estimate (421). That’s less difference than I would expect from intuition and experience but, again, the methods are rather opaque for me to be able to tell for sure. The implication is that Nmin may be too large and, therefore, the bycatch results would exceed PBR by even greater factors than those presented.

Open Science considerations

The Data Availability statement appears in conflict with the journal’s policy, which states that making data available only on request is not sufficient. I would also add that I didn’t see any indication that the analytical code would be made available. While this may not be a journal requirement, it is a common practice these days that does a lot to advance the principles of Open Science (e.g. https://www.cos.io/open-science ). Many readers would be grateful for it.

Other suggestions

I’ve marked quite a few suggestions and requests for clarification on the uploaded PDF of the manuscript.

I hope this review is helpful and I wish the authors all the best for publishing their findings on this important topic for the conservation of Saimaa seals. Sincerely,

Peter Boveng

2024-08-02

6. PLOS authors have the option to publish the peer review history of their article (what does this mean?). If published, this will include your full peer review and any attached files.

Reviewer #1: **Yes: **Brendan P. Kelly

Reviewer #2: **Yes: **Peter Boveng

---

## [Author Response · Author response to Decision Letter 0]

2 Sep 2024

Dear reviewers, 

Thank you for the very clear suggestions on how to improve the quality of the manuscript to meet the standards of PLOS ONE Journal. The file 'Response to Reviewers.docx' file includes revision information with reviewer comments (left), and author responses (right) in two side-by-side tables. The line numbers in this ‘Response to Reviewers.docx’ file refer to the line numbers in the ‘Revised Manuscript with Track Changes.docx’ file.

---

## [Editor Report · Decision Letter 1]

17 Sep 2024

Effects of fishing restrictions on the recovery of the endangered Saimaa ringed seal (Pusa hispida saimensis) population

PONE-D-24-24142R1

Dear Dr. Jounela,

We’re pleased to inform you that your manuscript has been judged scientifically suitable for publication and will be formally accepted for publication once it meets all outstanding technical requirements. I am sorry for the delay in providing this decision letter to you. As mentioned in my personal email to you, the journal does not permit acknowledgement of reviewers directly in the paper and they will receive an acknowledgement in the annual summary acknowledgement provided by the journal. I appreciate your work and that of the reviewers in helping to improve the quality of final paper, which I am pleased to accept.

Kind regards,

Lee W Cooper, Ph.D.

Section Editor

PLOS ONE

---

## [Editor Report · Acceptance letter]

29 Sep 2024

PONE-D-24-24142R1 

PLOS ONE

Dear Dr. Jounela, 

I'm pleased to inform you that your manuscript has been deemed suitable for publication in PLOS ONE. Congratulations! Your manuscript is now being handed over to our production team.

Kind regards, 

on behalf of

Dr. Lee W Cooper 

Section Editor

PLOS ONE